# Machine Learning for Bankruptcy Prediction in the American Stock Market: Dataset and Benchmarks

Gianfranco Lombardo [1,†], Mattia Pellegrino [1,†], George Adosoglou [2,†], Stefano Cagnoni [1,†], Panos M. Pardalos [2,†] and Agostino Poggi [1,*,†]

1. Department of Engineering and Architecture, University of Parma, 43124 Parma, Italy
2. Department of Industrial and Systems Engineering, University of Florida, Gainesville, FL 32611, USA
* Correspondence: agostino.poggi@unipr.it
† These authors contributed equally to this work.

**Abstract:** Predicting corporate bankruptcy is one of the fundamental tasks in credit risk assessment. In particular, since the 2007/2008 financial crisis, it has become a priority for most financial institutions, practitioners, and academics. The recent advancements in machine learning (ML) enabled the development of several models for bankruptcy prediction. The most challenging aspect of this task is dealing with the class imbalance due to the rarity of bankruptcy events in the real economy. Furthermore, a fair comparison in the literature is difficult to make because bankruptcy datasets are not publicly available and because studies often restrict their datasets to specific economic sectors and markets and/or time periods. In this work, we investigated the design and the application of different ML models to two different tasks related to default events: (a) estimating survival probabilities over time; (b) default prediction using time-series accounting data with different lengths. The entire dataset used for the experiments has been made available to the scientific community for further research and benchmarking purposes. The dataset pertains to 8262 different public companies listed on the American stock market between 1999 and 2018. Finally, in light of the results obtained, we critically discuss the most interesting metrics as proposed benchmarks for future studies.

**Keywords:** bankruptcy prediction; deep learning; multi-head; LSTM; machine learning; stock market

## 1. Introduction

Since the 2007/2008 financial crisis, most financial institutions, lenders, and academics have become interested in predicting corporate bankruptcy. Usually, corporate bankruptcy costs spread to the whole economy, resulting in cascade effects that impact many companies [1,2].

Despite that different research works have already demonstrated the ability of machine learning (ML) to assess the likelihood of companies' default, making a fair comparison among all the proposed approaches in the literature remains challenging for several reasons: (a) most of the datasets are not publicly available or are only related to specific economic scenarios like private companies in different countries [3,4]. For private companies, little information is generally available, which makes it difficult to exploit other sources of information that may improve bankruptcy prediction performance (e.g., textual disclosures [5], annual reports [6], stock market data) and that can be used by more complex models; (b) bankruptcy prediction actually involves different tasks: the default prediction in tasks for the next year, using past data, and the survival probability prediction task that aims to predict the probability that a company will face financial distress in *k* years. Most datasets cannot permit the performing of both tasks, and this is a clear limitation to the development of intelligent models that aim to generalize; (c) bankruptcy prediction models are usually trained on imbalanced data including few examples of the bankruptcy class: there is still no general accepted metric to assess bankruptcy prediction performance with machine learning. Indeed, the prediction accuracy can be misleading since it gives the same cost to false positives and false negatives

while making, for example, the wrong prediction on a company that is going to go bankrupt has in general a higher cost than the former.

At the same time, as a means of addressing class imbalance, the common metric reported for bankruptcy prediction is the Area Under the Curve (AUC). This, however, can also be misleading since it refers to the overall performance of a classifier without providing details regarding the performance of each class or the confusion matrix.

In this research work, we investigated the bankruptcy prediction with machine learning on the American stock market, which represents one of the most important drivers of the world economy. The reasons for this choice are summarized in the following: (a) different kinds of data are available for public companies in the American market: stock prices, accounting variables, and annual and quarterly financial reports. It allows the dataset to be improved in the future; (b) a corporate default in the American stock market can have a large impact on the overall economy.

The main contributions of our paper are the following:

1. We collected a dataset for bankruptcy prediction considering 8262 different companies in the stock market in the time interval between 1999 and 2018. The dataset has been made public (https://github.com/sowide/bankruptcy_dataset, accessed on 14 August 2022) for the scientific community for further investigations as a benchmark and thus it can be easily enriched with data coming from other sources pertaining to the same companies. Our dataset faithfully followed the FAIR Data Principles [7]:

   (a) Findable: our data is indexed in a searchable resource and had a unique and persistent identifier.
   (b) Accessible: our data is understandable to humans and machines and it is deposited in a trusted repository.
   (c) Interoperable: we used a formal, accessible, shared, and broadly applicable language for knowledge representation.
   (d) Reusable: we provided accurate information on provenance and clear usage licenses.

2. We investigated two different bankruptcy prediction tasks: The first task (T1) is the default prediction task which aims to predict the company status in the next year using time series of accounting data or just the last fiscal year available. The second task (T2) is the survival probability prediction task in which the model tries to predict the company status over $k$ years in the future.

3. In light of the results achieved, we critically discuss the most interesting metrics as proposed benchmarks for the future, such as: Area Under the Curve (AUC), precision, recall (sensitivity), type I error, type II error, and macro- and micro-F1 scores for each class.

The paper is organized as follows: In Section 2, we provide an overview the state-of-the-art approaches for bankruptcy prediction. In Section 3, we describe in detail the dataset that has been used for this study. In Section 4, we review and describe all the machine-learning algorithms that we used in our experiments. In Section 5, we introduce the metrics used for the imbalanced scenario encountered in this study. In Section 6, we describe the first task we evaluated in this work, concerning the prediction of a company's health status based only on data from previous years. In Section 7, we present our second task where we performed a survival probability task on companies within the dataset. In Section 8, we show and describe the experimental results, and finally in Section 9 we summarize the results with a critical discussion about the metrics.

## 2. Related Works

The recent advancements in machine learning (ML) led to new, innovative, and functional approaches [8,9]. Moreover, they have enabled the development of intelligent models that try to assess the likelihood of companies' default by looking for relationships among different types of financial data, and the financial status of a firm in the future [10–17]. Different ML algorithms and techniques such as Support Vector Machine (SVM) [18], boost-



ing techniques [19], discriminant analysis [20], and Neural Networks [5] have been used in the literature for this task. Moreover, different architectures have been evaluated to identify effective decision boundaries for this binary classification problem, such as the least absolute shrinkage and selection operator [21], dynamic slacks-based model [22], and two-stage classification [13]. However, although default prediction models have been studied for decades, several issues remain. Interestingly, some new issues have even been introduced with the recently increased exploitation of machine-learning models. Indeed, since the Z-Score model was proposed by Altman in 1968 [23], research mainly focused on accounting-based ratios as markers to detect and understand if a firm is likely to face financial difficulties, such as bankruptcy. Scoring-based models use discriminant analysis to provide ordinal rankings of default risk but are often computed from small datasets using statistical and probabilistic models that focus more on explainability and explicability but miss generalization over time and across different sectors [24]. Other examples are the Kralicek quick test [25] and the Taffler model [26].

In [27], a step towards modern machine learning was made by introducing a binary response model that uses explanatory variables and applies a logistic function for bankruptcy prediction [28]. However, the main goal of these models' class is not to identify a decision boundary in the feature space but only to select a decision based on an output threshold that was statistically significant in the past for the specific sector. For example, Altman suggested two thresholds, 1.81 and 2.99.

Specifically, an Altman's Z-score above the 2.99 threshold means that firms are not expected to default in the next two years, below 1.81 that they are expected to default, while the interval between the two thresholds is named the "zone of ignorance" where no clear decision can be taken. However, even though many practitioners use this threshold, in Altman's view, this is an unfortunate practice since over the past 50 years, credit-worthiness dynamics and trends have changed so dramatically that the original zone cutoffs are no longer relevant [24].

Moreover, we still lack a definite theory for the bankruptcy prediction task [18,29] and in particular, a generally accepted performance metric is missing along with a formal theoretical framework. As a consequence, the most common methodology in bankruptcy prediction tasks is identifying discriminant features using a trial and error approach with various accounting-based ratios [15,16].

Machine-learning models usually need large datasets to be trained and suffer when class imbalance is strong as in bankruptcy, since default events are quite rare. Learning from imbalanced data requires dealing with several challenges, especially when the most important class that should be recognized is exactly the one that is least represented in the dataset. This issue is strongly related to the lack of a general performance metric.

Machine-learning techniques like ensemble methods were firstly explored for default prediction by Nanni et al. [30]. Kim et al. showed a much better performance for the ensembles compared to standalone classifiers, while their results were also confirmed by Kim et al. [31]. Wang et al. further analyzed the performance of ensemble models, finding that bagging outperformed boosting for all credit databases in terms of average accuracy, as well as type I and type II error [32]. In [33], Barboza et al. show that, on average, machine-learning models exhibit 10% more accuracy than scoring-based ones. Specifically, in this study, Support Vector Machines (SVM), Random Forests (RF) as well as bagging and boosting techniques were tested for predicting bankruptcy events and were compared with results from the discriminant analysis, Logistic Regression, and Neural Networks. The authors found that bagging, boosting, and RF outperformed all other models. However, since the dataset has not been released and the models' hyper-parameters are not reported, it is difficult to replicate such results and understand if the performance improves because of the quality of the models or because the authors take into account some other financial variables as features.

Considering that the comparisons of the models are still inconclusive, new studies exploring different models, contexts, and datasets are relevant. A firm's failure is likely

to be caused by difficulties experienced over time and not just the previous year. In light of this, the dynamic behavior of firms should be considered in a theoretical framework for bankruptcy prediction such as growth measures and changes in some variables [33]. Several research works investigated this aspect, but the results are again inconclusive and often not reproducible. In [28], the authors show that firms exhibit failure tendencies as much as five years prior to the actual event. On the other hand, Mossman et al. [34] pointed out that the models are only capable of predicting bankruptcy two years prior to the event, which improves to three years if used for multiperiod corporate default prediction [35]. In most studies, ratios are analyzed backward in time starting with the bankruptcy event and going back until the model becomes unreliable or inaccurate. Moreover, most of the bankruptcy prediction models in the literature do not take advantage of the sequential nature of the financial data. This lack of multi-period models is also emphasized in a literature review by Kim et al. [36].

## 3. Dataset

Since most of the bankruptcy models are evaluated on private datasets or small publicly available ones, we provide a novel dataset for bankruptcy prediction related to the public companies in the American stock market (New York Stock Exchange and NASDAQ). We collected accounting data from 8262 different companies in the period between 1999 and 2018. The stock market is dynamic with new companies becoming public every year, changing properties and names, or being removed or suspended from the market as a result of acquisitions or regulatory action. For this reason, we consider the same companies used in [6,37] since this set of firms has been proved to be a fair approximation of the American stock market for each year in that time interval. According to the Security Exchange Commission (SEC), a company in the American market is considered bankrupt in two cases:

- If the firm's management files Chapter 11 of the Bankruptcy Code to "reorganize" its business: management continues to run the day-to-day business operations but all significant business decisions must be approved by a bankruptcy court.
- If the firm's management files Chapter 7 of the Bankruptcy Code: the company stops all operations and goes completely out of business.

When these events occur we label the fiscal year before the chapter filing as "bankruptcy" (1) for the next year. Otherwise, the company is considered healthy (0). In light of this, our dataset enables learning how to predict bankruptcy at least one year before it happens and, as a consequence, it is possible to deal with the default prediction task using time series and also to deal with the survival probability task looking ahead. Figure 1 shows the percentage of companies' default for each year in the dataset. This value can be underestimated due to the exclusion of some companies in the past because of their small market capitalization, but it appears to agree with the literature that usually reports that only a percentage below 1% of the available firms in the market is likely to default every year under normal conditions. However, in some periods, bankruptcy rates have been higher than usual, for example during the Dot-com Bubble in the early 2000s and the Great Recession in 2007–2008. Our dataset distribution reflects this condition, see Table 1.

For all the companies and for each year, we selected 18 accounting and financial variables. Features were selected according to the most frequently used ratios, and accounting information to which the literature refers [21,23,38]. The dataset has no missing values or synthetic and imputed added values. Finally, the resulting dataset of 78,682 firm-year observations is divided into three subsets according to the period of time: a training set, a validation set, and a test set. We used data from 1999 until 2011 for training, data from 2012 until 2014 for validation and model comparison, and the remaining years from 2015 to 2018 as a test-set to prove the ability of the models to predict bankruptcy in real never seen cases and macroeconomic conditions. Table 2 shows the full list of the 18 features available in the dataset and their description.

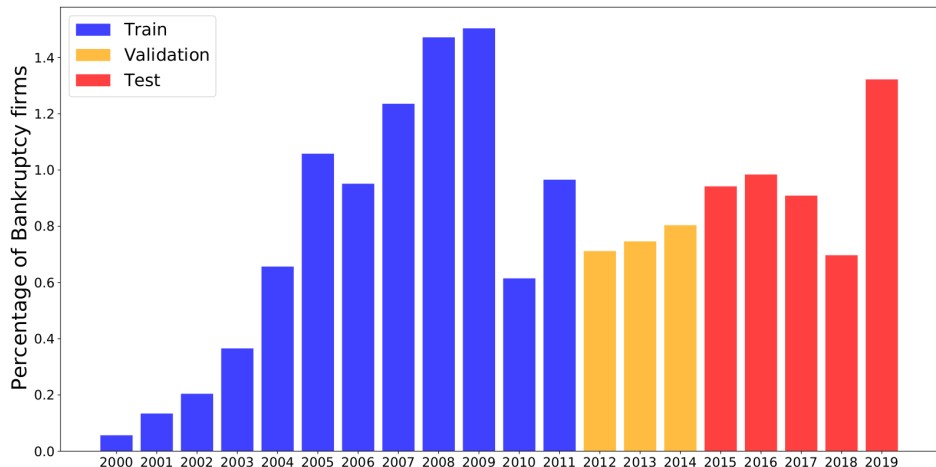

**Figure 1.** Rate of bankruptcy in the dataset (2000–2019) with financial variables in the period (1999–2018). The next subdivision in training, validation, and test is highlighted with different colors.

**Table 1.** The table provides the firm distribution by year in the dataset.

| Year | Total Firms | Bankrupt Firms | Year | Total Firms | Bankrupt Firms |
|------|-------------|----------------|------|-------------|----------------|
| 2000 | 5308 | 3 | 2010 | 3743 | 23 |
| 2001 | 5226 | 7 | 2011 | 3625 | 35 |
| 2002 | 4897 | 10 | 2012 | 3513 | 25 |
| 2003 | 4651 | 17 | 2013 | 3485 | 26 |
| 2004 | 4417 | 29 | 2014 | 3484 | 28 |
| 2005 | 4348 | 46 | 2015 | 3504 | 33 |
| 2006 | 4205 | 40 | 2016 | 3354 | 33 |
| 2007 | 4128 | 51 | 2017 | 3191 | 29 |
| 2008 | 4009 | 59 | 2018 | 3014 | 21 |
| 2009 | 3857 | 58 | 2019 | 2723 | 36 |

**Table 2.** The 18 numerical bankruptcy features we considered in our tests.

| | Variable Name | Description |
|------|---------------|-------------|
| **X1** | Current assets | All the assets of a company that are expected to be sold or used as a result of standard business operations over the next year |
| **X2** | Cost of goods sold | The total amount a company paid as a cost directly related to the sale of products |
| **X3** | Depreciation and amortization | Depreciation refers to the loss of value of a tangible fixed asset over time (such as property. machinery, buildings, and plant). Amortization refers to the loss of value of intangible assets over time. |
| **X4** | EBITDA | Earnings before interest,taxes, depreciation and amortization: Measure of a company's overall financial performance alternative to the net income |
| **X5** | Inventory | The accounting of items and raw materials that a company either uses in production or sells. |

**Table 2.** *Cont.*

|     | Variable Name | Description |
| --- | --- | --- |
| **X6** | Net Income | The overall profitability of a company after all expenses and costs have been deducted from total revenue. |
| **X7** | Total Receivables | The balance of money due to a firm for goods or services delivered or used but not yet paid for by customers. |
| **X8** | Market value | The price of an asset in a marketplace. In our dataset it refers to the market capitalization since companies are publicly traded in the stock market |
| **X9** | Net sales | The sum of a company's gross sales minus its returns, allowances, and discounts. |
| **X10** | Total assets | All the assets, or items of value, a business owns |
| **X11** | Total Long term debt | A company's loans and other liabilities that will not become due within one year of the balance sheet date |
| **X12** | EBIT | Earnings before interest and taxes |
| **X13** | Gross Profit | The profit a business makes after subtracting all the costs that are related to manufacturing and selling its products or services |
| **X14** | Total Current Liabilities | It is the sum of accounts payable, accrued liabilities and taxes such as Bonds payable at the end of the year, salaries and commissions remaining |
| **X15** | Retained Earnings | The amount of profit a company has left over after paying all its direct costs, indirect costs, income taxes and its dividends to shareholders |
| **X16** | Total Revenue | The amount of income that a business has made from all sales before subtracting expenses. It may include interest and dividends from investments |
| **X17** | Total Liabilities | The combined debts and obligations that the company owes to outside parties |
| **X18** | Total Operating Expenses | The expense a business incurs through its normal business operations |

In light of this, the dataset could be used to build and validate different ML models for both of the main two tasks in bankruptcy prediction we show in this research work. Moreover, since the dataset has a temporal dimension, several time series analysis techniques could be exploited as long as there are unsupervised methodologies.

## 4. Machine-Learning Models

In this section, we briefly review and describe all the machine-learning algorithms we used for the experiments described in the next sections.

### 4.1. Support Vector Machine

Support Vector Machine is one of the oldest ML algorithms and aims to identify the decision boundaries as the maximum-margin hyperplane separating two classes. The hyperplane equation is given by Equation (1).

$$f(x) = \omega^T \cdot x + b \tag{1}$$

where $\omega$ is the normal vector and $b$ the bias. The objective function of SVM can be expressed as:

$$\begin{cases} \min\left\{\frac{1}{2}\|\omega\|^2\right\} \\ \text{s.t.} \begin{cases} y_i - \omega^T x_i - b \leqslant \varepsilon \\ \omega^T x_i + b - y \leqslant \varepsilon \end{cases} \end{cases} \tag{2}$$

where $\epsilon$ is the deviation between $f(x)$ and the target $y_i$.

### 4.2. Random Forest

Random Forest is an ensemble learning algorithm developed by Breiman [39]. Ensemble learning is a way to combine different basic classifiers ("weak" classifiers) to compose a new one (strong learner), more complex, more efficient, and more precise. The weak classifiers should make independent errors in their predictions, and thus a strong classifier can be composed of different algorithms or if the same algorithm is used the models should be trained with different subsets of the training set.

Random Forest is an ensemble bagging tree-based learning algorithm. In particular, the Random Forest Classifier is a set of decision trees that are trained using randomly selected subsets of the training set and randomly selected subsets of features.

A Random Forest Classifier is composed of a collection of classification trees $h$ (estimators):

$$\{h(x, T, \Theta_k), k = 1, 2, \ldots, K\} \tag{3}$$

where $(\Theta_k)$ represents identically and independently distributed random vectors, and each tree casts a unit vote for the most likely class at input $x$. Each tree in the collection votes (only once) to assign the sample to a class, considering the $x$ feature vector. The final choice is to attribute the example to the class that obtained the majority of votes. A graphic representation of the algorithm is presented in Figure 2.

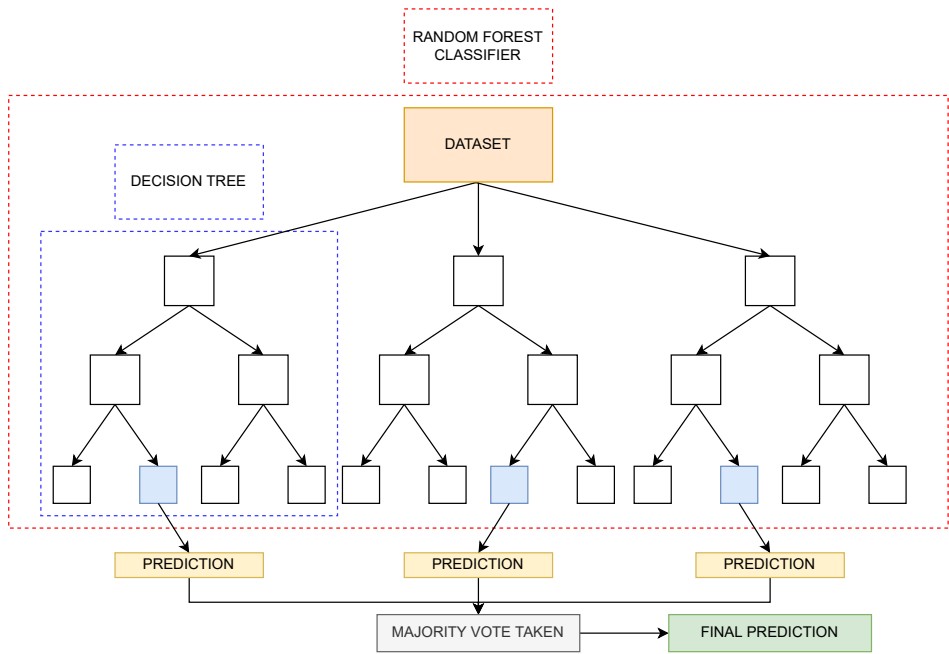

**Figure 2.** Graphic representation of Random Forest algorithm for classification.

### 4.3. Boosting Algorithms

Boosting is a subset of ensemble methods where a collection of models are trained sequentially to permit every model to improve and compensate for the weakness of its predecessor.

Boosting algorithms differ in how they create and aggregate weak learners during the sequential stacking process. In our work, we used various boosting algorithms:

- **AdaBoost** [40]: This was the first boosting algorithm developed for classification and regression tasks. It fits a sequence of weak learners on different weighted training data. It gives incorrect predictions more weight in sub-sequence iterations and less weight to correct predictions. In this way, it forces the algorithm to "focus" on observations that are harder to predict. The final prediction comes from weighing the majority vote or sum.

  The algorithm begins by forecasting the original dataset and giving the same weight to each observation. If the prediction is incorrect, using the first "weak" learner, the algorithm will give a higher weight value to that observation. This procedure is iterated until the model reaches a predefined value of accuracy.

  AdaBoost is typically easy to use because it does not need complex parameters during its tuning procedures and it shows low sensitivity to overfitting. Moreover, it is able to learn from a small set of features learning incrementally new information. However, Adaboost is sensitive to noisy data and abnormal values.

- **Gradient Boosting** [41]: This algorithm uses a set of weak predictive models, typically decision trees. Gradient Boosting trains many models sequentially that are then composed using the additive modeling property. In each training epoch, a new learner is added to increase the accuracy of the previous one. Each model gradually minimizes the whole system loss function using the Gradient Descent algorithm.

- **XGBoost** (Extreme Gradient Boosting) [42]: XGBoost is an optimized distributed Gradient Boosting library designed to be highly efficient, flexible, and portable. XGBoost minimizes a regularized (L1 and L2) objective function that combines a convex loss function based on the difference between the predicted and target outputs and a penalty term for model complexity. The training proceeds by adding new trees that predict the residuals or errors of prior trees that are then combined with previous trees to make the final prediction (4).

$$F_m(X) = F_{m-1}(X) + \alpha_m h_m(X, r_{m-1}), \tag{4}$$

where $\alpha_i$ and $r_i$ are the regularization parameters and residuals computed with the $i$-th tree, respectively, and $h_i$ is a function that is trained to predict residuals, $r_i$ using $X$ for the $i$-th tree.

$\alpha_i$ is computed using the residuals, $r_i$ solving the following optimization problem:

$$\arg \min_{\alpha} = \sum_{i=1}^{m} L(Y_i, F_{i-1}(X_i) + \alpha h_i(X_i, r_{i-1})) \tag{5}$$

where $L(Y, F(X))$ is a differentiable loss function.

### 4.4. Logistic Regression

Binary Logistic Regression models the relationship between a set of independent variables and a binary dependent variable. The goal is to find the best model that describes the relationship between the dependent variable and multiple independent variables.

The Logistic Regression's dependent variable could be binary or categorical and the independent ones could be a mixture of continuous, categorical, and binary variables.

The general form of Logistic Regression is as follows:

$$z = a + b_1 x_1 + b_2 x_2 + b_3 x_3 + \ldots + b_m x_m \tag{6}$$

$$y = \frac{1}{(1 + e^{-z})} \tag{7}$$

where $x_1, x_2, \ldots, x_m$ is the feature vector and $z$ is a linear combination function of the features. The parameters $b_1, b_2, \ldots, b_m$ are the regression coefficients to be estimated. The output is between 0 and 1, and, usually, if the output is above the threshold of 0.5 the model predicts class 1 (positive) and otherwise class 0 (negative).

### 4.5. Artificial Neural Network

An Artificial Neural Network (ANN) is a non-linear function approximator. It consists of an input layer of neurons and an unspecified number of hidden layers and a final output layer. Each neuron performs a weighted sum of its inputs, and finally, it applies an activation function that determines the output of each neuron. When the activation function is a Sigmoid function, the single neuron works as a Logistic Regression without the classification threshold. Figure 3 shows a general structure of an ANN with the input layer, two hidden layers, and the final output layer whose structure strongly depends on the task it should perform. The main common architecture is the feed-forward ANN where each neuron is linked to every neuron in the next layer but without exhibiting any intra-layer connection among neurons belonging to the same layer. Each layer can be seen as a partial approximation of the final function.

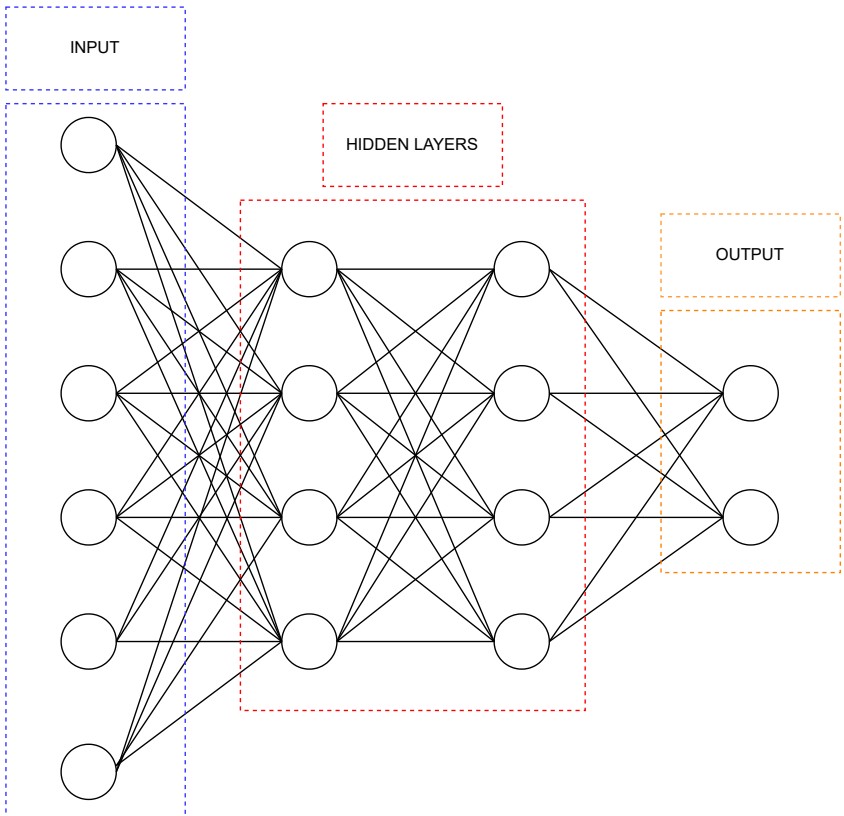

**Figure 3.** Representation of a generic Neural Network. Nodes are called *neurons* and the links are called *weights*.

Each connection has a *weight* $\omega$ assigned that is randomly initialized at the beginning. The output $h_i$, of a neuron $i$, in the hidden layer, is:

$$h_i = \sigma(\sum_{j=1}^{N} \omega_{ij} x_j + b_i) \tag{8}$$

where $\sigma()$ is the activation function, $N$ the number of input neurons of the layer, $\omega_{ij}$ the weights, $x_j$ inputs of the neurons, and $b_i$ the bias terms of the hidden neurons.

The goal of the activation function is to bound the value of the neuron so that the Neural Network is not stuck by divergent neurons. Weight estimation for each connection is the main goal of the ANN's training. This step is usually performed using the back-propagation algorithm [43] that minimizes an objective function that measures the distance between the desired and actual output of the network. Inputs and outputs from a Neural

Network can be binary or even symbols when data are appropriately encoded. This feature confers a wide range of applicability to Neural Networks.

### 5. Metrics for Imbalanced Bankruptcy Prediction Tasks

The two bankruptcy prediction tasks we propose in the following sections have been implemented as binary prediction tasks where the positive class (1) indicates bankruptcy while the negative class (0) means that a company has been classified as healthy. To compare our models and investigate their effectiveness, we used different metrics considering the imbalance condition in the validation and the test sets. These metrics will be critically discussed in Section 8 in light of our results for the two tasks. According to the binary classification task we used to formulate the bankruptcy prediction problem, the following variables represent:

- **True positive (TP)**: The number of bankrupted companies that have been correctly predicted as such.
- **False negative (FN)**: The number of bankrupted companies that have been wrongly predicted as healthy firms.
- **True negative (FN)**: The number of actually healthy companies that have been correctly predicted as such.
- **False positive (FP)**: The number of healthy companies that have been wrongly predicted as bankrupted by the model.

Since the validation and test sets are both imbalanced for both tasks with a prevalence of healthy companies ($\sim$96–97%), we did not compare the models in terms of accuracy of the model. Indeed, the proportion of correct matches would be insufficient in assessing the model performance. We firstly use the Area Under the Curve (AUC) for all the comparisons as it is commonly used in the literature to compare the performance of models on imbalanced datasets and specifically to evaluate the bankruptcy models in general. AUC measures the ability of a classifier to distinguish between classes and is used as a summary of the Receiver Operating Characteristic (ROC) curve. The ROC curve is created by plotting the true positive rate (TPR) against the false positive rate (FPR) at various threshold settings.

In addition, we investigated other important metrics that can be used to clarify the models' performance depending on the target stakeholders. These are the precision, recall, and $F_1$ scores for each class. The precision achieved for a class is the accuracy of that class' predictions. The recall (sensitivity) is the ratio of the class instances that are correctly detected by the classifier. The $F_1$ score is the harmonic mean of precision and recall: whereas the regular mean treats all values equally, the harmonic one gives more weight to low values. As a consequence, a high $F_1$ score for a certain class is achieved only if both its precision and recall are high. Equations (11)–(13) report how these quantities are computed for the positive class. The definitions for the negative class are exactly the same by inverting positives with negatives.

$$Precision = \frac{TP}{(TP + FP)} \tag{9}$$

$$Recall = \frac{TP}{(TP + FN)} \tag{10}$$

$$F_1 score = \frac{2}{\frac{1}{Precision} + \frac{1}{Recall}} \tag{11}$$

Moreover, we computed and reported two other global metrics for the classifier that have been selected because they enable an overall evaluation of the classifier on both classes:

- **The macro-$F_1$ score**: The macro-$F_1$ score is computed as the arithmetic mean of the $F_1$ score of all the classes.
- **The micro-$F_1$ score**: It is used to assess the quality of multi-label binary problems. It measures the $F_1$ score of the aggregated contributions of all classes.

Finally, we used two other metrics that are often evaluated in the bankruptcy prediction models. Because bankruptcy is a rare event, using the classification accuracy to measure a model's performance is misleading since it assumes that *type I error* (Equation (12)) and *type II error* (Equation (13)) are equally costly. Actually, the cost of false negatives is much higher than that of false positives for a financial institution. An error has a different cost depending on the class that has been incorrectly predicted. The cost of predicting a company going into default as healthy is much higher than the cost of predicting a company that will default as healthy. In light of this, we explicitly computed and reported type I and type II errors and compared the classifiers focusing in particular on type II and recall of the positive class.

$$Type \quad I \quad error = \frac{FP}{TN + FP} \tag{12}$$

$$Type \quad II \quad error = \frac{FN}{TP + FN} \tag{13}$$

## 6. Task T1: Default Prediction with Historical Time Series

The first task we performed using the dataset is training each ML model presented in Section 4 for default prediction with historical time-series accounting data. Our first step was to perform the most classical task of predicting a company's health status for the next year based solely on data from the previous year. Furthermore, we attempted to answer the open question in the literature regarding the number of years that should be considered in order to maximize the performance of the bankruptcy prediction model. To achieve this, we define a Window Length (WL) variable that refers to the number of fiscal years considered as a temporal window for the prediction. We trained all the models using data between 1999 and 2011, and we made the first comparison using the Validation set (2012–2014) in terms of AUC.

Finally, we report the results of the test set using the best models identified on the validation set. This last step is necessary to verify the ability of the models to generalize. The average number of years available for each company in the dataset is 8 years. However, we evaluated a WL ranging from one to five years for two main reasons:

- Five years is the general maximum number of years found in the literature to be useful.
- When increasing the WL considered, the most recent companies are excluded because they do not have available data. In general, considering more years leads to smaller training and test sets. This could introduce a statistical bias, causing the analysis to focus on only the more mature and stable companies that have existed for several years while ignoring the relatively newer, smaller companies that are riskier and have higher default rates. This could introduce a statistical bias forcing the analysis to consider only the more structured and stable companies that have existed for several years while ignoring the relatively new companies with smaller market capitalization and which are thus riskier and have a higher probability of default, particularly during periods of economic decline.

Figure 4 shows the distribution of the two classes of bankruptcy and healthy samples for the training, test, and validation by selecting different Window Lengths (WLs).

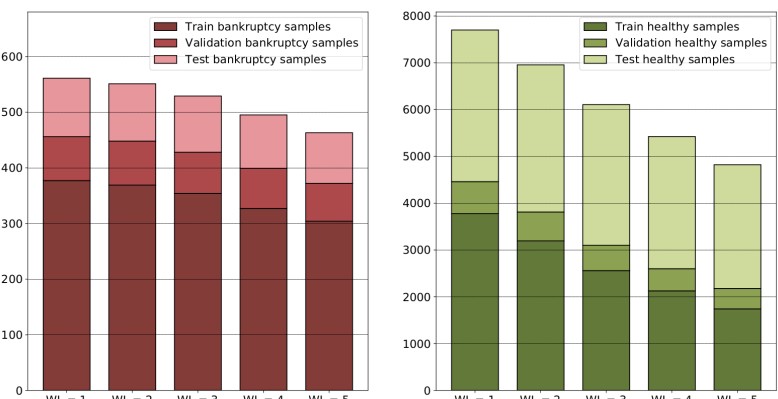

**Figure 4.** Bankruptcy and healthy sample distribution in the training (2000–2011), validation (2012–2014), and test (2015–2018) sets by varying the length of the window (WL) considered.

*6.1. Models Comparison and Selection for Default Prediction*

All the models were trained using the same training set (1999–2011) and compared using the same validation set (2012–2014). However, learning from an imbalanced training set led to unsatisfactory results for the bankruptcy class in the validation set. For this reason, we decided to use a random balanced training set that is evaluated for different runs. Indeed, every model is evaluated over 100 independent and different runs: for every run, the training set is balanced with exactly the same number of bankruptcy examples and a random choice of healthy examples from the same period. The number of features changes according to the number of available variables for the temporal window length selected.

A binary classification task is implemented in each model, where the positive class (1) represents a bankruptcy case in the next year, and the negative class (0) represents a healthy case. For RF, AB, GB, and XGB, we used 500 estimators for a fair comparison, while the other specific parameters are the default ones provided in the Scikit-Learn implementations. The ANN is a multi-layer perceptron with three layers. In the first layer, the number of neurons is equal to the number of input features. The hidden layer has half as many neurons as the first layer, and finally, the output layer uses a Sigmoid function to produce a binary prediction. All neurons except for those in the output layer use the Rectified Linear Unit activation (RELU) function. Each Neural Network is trained over 1000 epochs using the early-stopping regularization technique that prevents overfitting when the validation loss tends to increase for a patience number of times.

We compared all the models using the average Area Under the Curve (*AUC*) over 100 runs. Figure 5 reports the results we achieved trying to predict bankruptcy in the most common setting that exploits only the accounting variables of the last available fiscal year. On average, the ensemble tree-based ML models perform better than the others except for Adaboost. The best average result is achieved with the Random Forest (*AUC* = 0.748), although the Neural Network achieves the best *AUC* in a single run (*AUC* = 0.856). However, the ANN exhibits the highest variability in the results, which led to a lower average AUC (*AUC* = 0.653). Figure 6 reports the comparison among the models when using more than one year from the prediction (WL from two to five years). As a result of this experiment, we can assert that on average increasing the number of fiscal years in the input does not seem to impact the average AUC significantly:

- For WL = 2, best average AUC = 0.75 with Random Forest
- For WL = 3, best average AUC = 0.72 with Random Forest and Gradient Boosting
- For WL = 4, best average AUC = 0.74 with XGBoost
- For WL = 5, best average AUC = 0.754 with Random Forest

The highest average value is reached by Random Forest using a window length of 5 years. However, as in the first case, in general, the ensemble models exhibit better performance. However, the best absolute result is reached by the Neural Network again with *AUC* = 0.85. It should be highlighted that the Neural Network's high variability

concerning the results can be imputed to the random weights' initialization for every run. Probably performing several runs over the same training set could reduce this problem. Finally, Table 3 shows the variance for each model considering every window length.

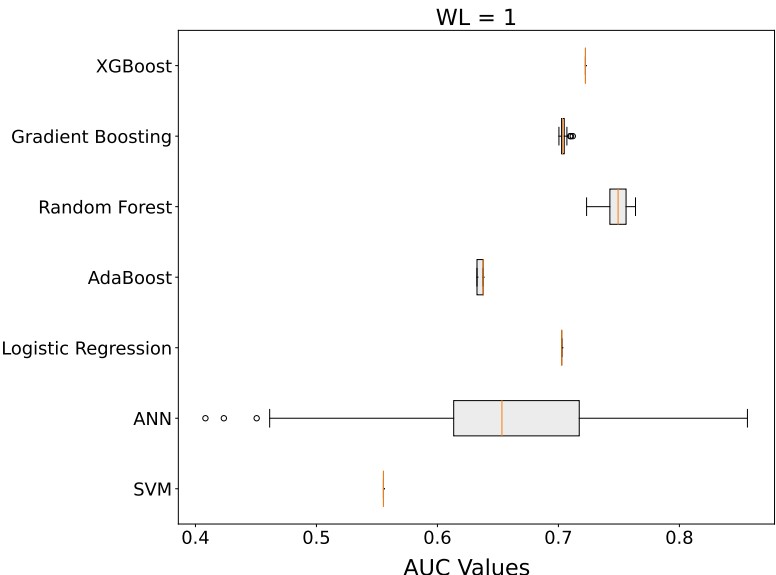

**Figure 5.** The box plot shows the locality, spread, and skewness groups of AUC values through their quartiles achieved for each model in 100 different runs with a different balance training set. The reported AUC values (horizontal axis) are related to the case of WL = 1 where all models use only variables from the last year. The lines (whiskers) extending from the boxes indicate the variability outside the upper and lower quartiles. The orange line represents the median between the first quartile and the third quartile. The circular points (fliers) are those located past the end of the whiskers.

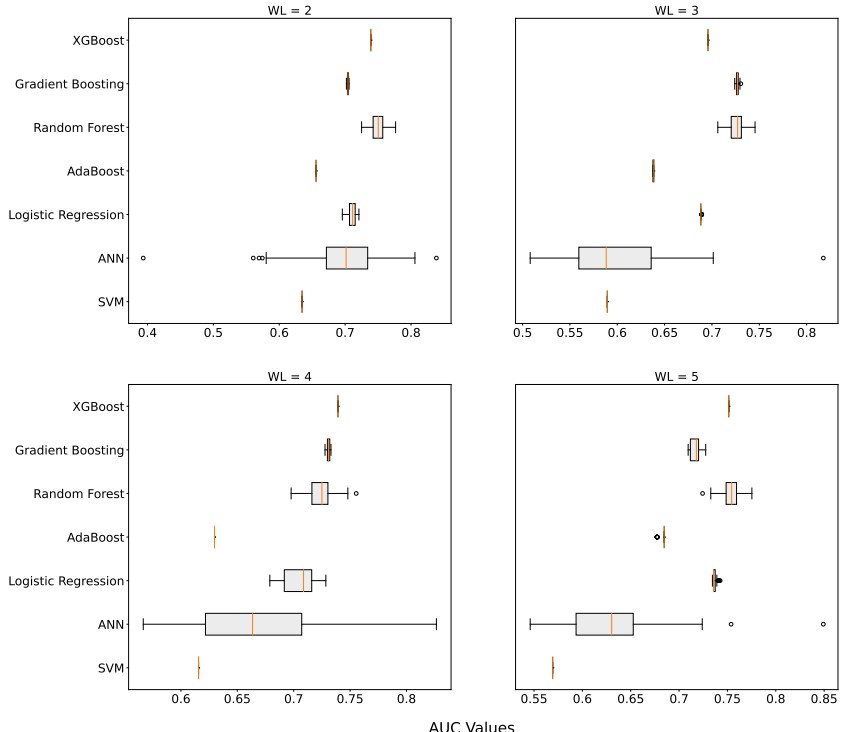

**Figure 6.** The four box-plots show the locality, spread, and skewness groups of AUC values achieved on the validation set for each model in the same conditions as for the first case by varying the WL parameter.

**Table 3.** Variance on the AUC values calculated for each model on the validation set for T2 task (LAW) and for T1 task (WL).

|  | LAW = 2 | LAW = 3 | LAW = 4 | LAW = 5 | WL = 1 | WL = 2 | WL = 3 | WL = 4 | WL = 5 |
|---|---|---|---|---|---|---|---|---|---|
| **Svm** | 4.930e-32 | 0.0 | 0.0 | 4.930e-32 | 0.0 | 1.233e-32 | 0.0 | 0.0 | 0.0 |
| **Ann** | 0.009 | 0.0075 | 0.0089 | 0.0103 | 0.0076 | 0.0037 | 0.0026 | 0.0029 | 0.0022 |
| **Logistic Regression** | 1.233e-32 | 1.232e-32 | 6.145e-07 | 2.488e-07 | 1.233e-32 | 2.831e-05 | 5.612e-08 | 0.00019 | 2.427e-06 |
| **AdaBoost** | 6.416e-07 | 0.0 | 8.430e-07 | 4.930e-32 | 6.559e-06 | 1.233e-32 | 2.039e-07 | 1.233e-32 | 6.951e-06 |
| **Random Forest** | 8.301e-05 | 8.208e-05 | 6.273e-05 | 9.655e-05 | 8.076e-05 | 0.00012 | 7.139e-05 | 0.00011 | 9.349e-05 |
| **Gradient Boosting** | 2.887e-05 | 1.616e-05 | 4.583e-06 | 2.462e-05 | 3.135e-06 | 8.098e-07 | 1.801e-06 | 1.445e-06 | 2.860e-05 |
| **XgBoost** | 4.930e-32 | 0.0 | 0.0 | 1.233e-32 | 4.930e-32 | 1.233e-32 | 1.233e-32 | 4.930e-32 | 4.930e-32 |

### 6.2. Results for Default Prediction

Once we compared the algorithms with 100 different runs on the validation set, ranked the average AUC achieved for every temporal window, and selected the best model on the validation set, we performed a final evaluation on the test set (2015–2018). These data have never been used for model training or comparison. It is important to evaluate on a different test set in terms of temporal period because it refers to a different economic cycle. Since we trained on data until 2011 that include the 2007/2008 sub-prime crisis and the European debt crisis in 2010–2011, there could be a bias concerning the knowledge learnt by the models about what is a company that is likely to go into bankruptcy. Evaluating all the models only on the validation set is not enough since we used that dataset to select the hyper-parameters. If a model performs well on both validation (2012–2014) and on the never used test set (2015–2018), we could asses that it is effectively able to generalize concerning bankruptcy companies. The best models on the validation set are in order Random Forest, Gradient Boosting, XGBoost, and Logistic Regression. Figure 7 presents the final results we achieved for this task on the American stock market. We also added the results achieved by the Neural Network with the best model that actually outperforms all the other machine-learning techniques in terms of AUC. It appears that this result contradicts what has been reported in the current literature, which is that bagging and boosting ensemble models perform better. The results achieved on the never seen test set by the ANN demonstrate what is currently known for other machine-learning applications: When a Neural Network is properly designed, trained, and fine-tuned with higher computational costs and when the best model is identified with random and grid searches, then Neural Networks usually outperform all the machine-learning baselines thanks to their ability to deal with non-linear dependencies.

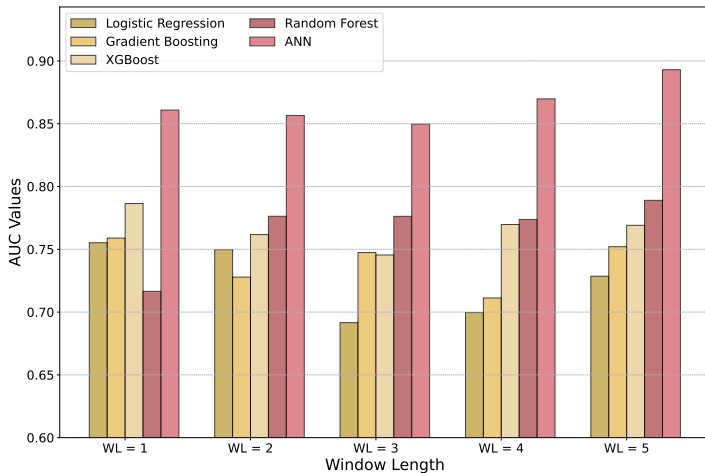

**Figure 7.** AUCs achieved by the best models on the test set (2015–2018) for task T1 using the best models selected on the validation set (2012–2014). The ANN is also considered because, although it yields the worst average AUC on the validation set, it achieves the best absolute performance on the test set when using the best models found on the validation set.

### 7. Task T2: Survival Probability Task

The second task we performed using the dataset is the years anticipated prediction of the bankruptcy events. This task is implemented by considering the event in the dataset and looking ahead a number of years; we named it the Look-Ahead Window (LAW). In practice, default prediction with WL = 1 is exactly the same as the one with LAW = 1. For this reason we investigated the LAW parameter ranging from two to five years. The main difference between task T1 and this task is that for task T2 the models exploit only a single year of accounting variables in the past depending on the LAW parameter. For example if LAW is selected equal to three years, it means that the model learns how to predict the companies' status looking over three years. This way of predicting bankruptcy is usually adopted to estimate the survival probability of a company within some years. All the experiments are conducted with the same methodological framework as for task T1. We trained all the models for 100 different runs by randomly undersampling the number of healthy examples in the training set in order to have a balanced set. We compared all the models on the validation set to select the best ones by varying the LAW parameter. Models settings are the same as for task T1. Figure 8 shows the models comparison. We remind readers that the validation set is unbalanced and this is the reason why the performances are measured in terms of the AUC. The results are similar to task T1. Adaboost achieves the worst performance on average along with Support Vector Machine for all the LAW parameters. The other ensemble models seem to outperform the other models and exhibit a small variance by changing the training set for each run. The Neural Network achieves the best absolute result for each window but presents a high performance variance. However, one should highlight that, in general, all the models achieve a better average AUC for the survival probability task rather than on the classical bankruptcy prediction (task T1). Table 3 shows the variance calculated on the AUC values on the validation set for each look-ahead window.

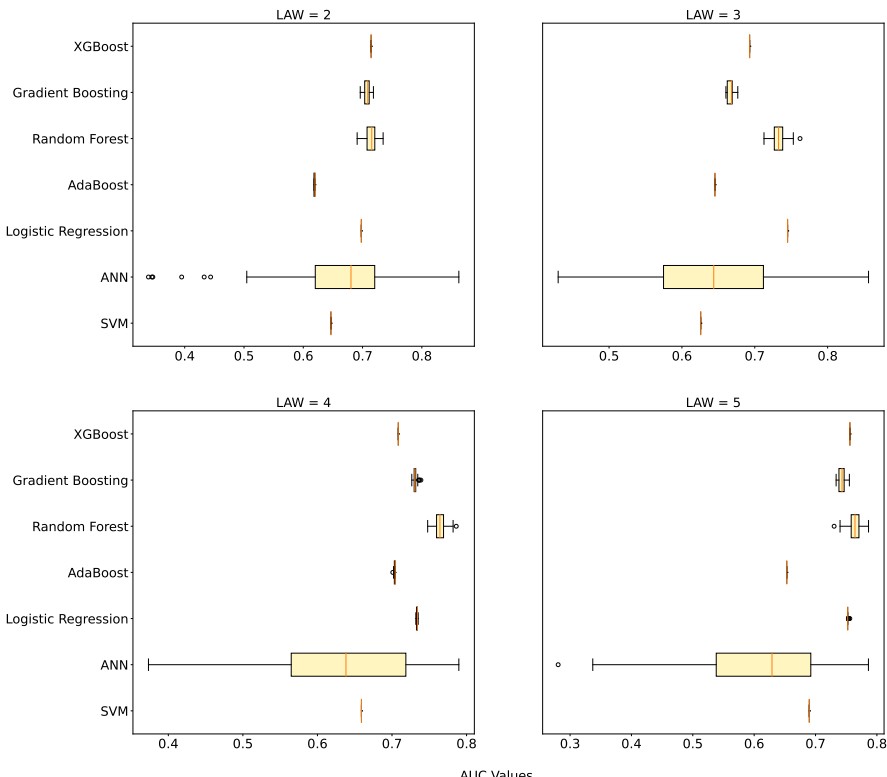

**Figure 8.** The four box plots show the locality, spread, and skewness groups of AUC values achieved for each model on the validation set for the survival probability task (T2) when the Look-Ahead Window (LAW) is changed between 2 and 5 years.

*Results for the Survival Probability Prediction*

As already carried out for task T1, we selected the best models on the validation set, and we measured the generalization ability of the models on the test set (2015–2018). We compared GB, XGB, LR, and RF. As expected from the previous models' comparison, the best model is Random Forest also in this case. However, also for this task, the best ANN model found on the validation set definitely outperforms all the other models, showing an additional demonstration of the better ability of this category of models to achieve better performance when properly designed. In general, the results of both validation and test experiments suggest that machine learning can better predict a company's status over a long period (from 3 to 5 years before) rather than over a short period (1 or 2 years). Results are presented in Figure 9.

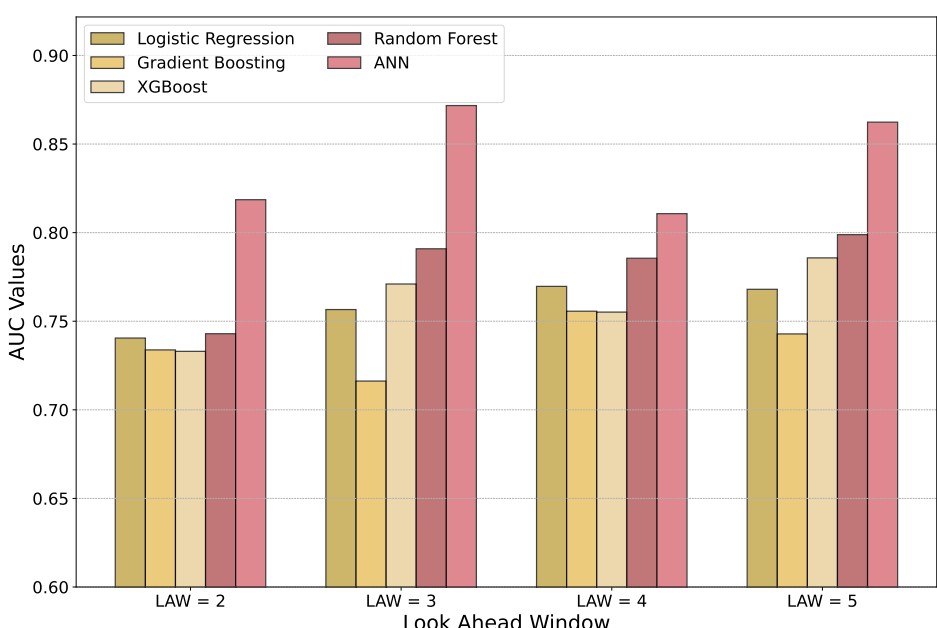

**Figure 9.** AUCs achieved by the best models on the test set (2015–2018) for task T2 (survival probability over time) prediction using the best models selected on the validation set (2012–2014).

## 8. Results

In this section, we present a deeper analysis of the results for both tasks T1 and T2 by also discussing the other metrics available. In Sections 6 and 7, we compared all the models in terms of the AUC, and we identified the best model for both tasks using this metric. This metric was chosen for two reasons: first, due to the imbalance test set, but also because it is most commonly used in literature for this task. Now, we would like to analyze the same results by looking at the other metrics in order to find some robust answers about the models for the two tasks. Tables 4 and 5 show all the results we achieved with all the models on the test set (2015–2018).

We show precision and recall for both the bankruptcy class and the healthy one. Starting from these, we computed the micro- and macro-$F_1$ scores for the overall classifier and finally the type I and type II errors. All the results were computed for all the temporal window WLs and LAWs of the two tasks.

In the end, we reported the complete confusion matrix of the experiments that we have never seen reported in the other papers because we strongly believe that this may help for correct comparisons and usage as benchmarks for future investigations.

**Table 4.** Overall results on the test set (2015–2018) for the T1 task: default predictions with historical data. Rec indicates the recall and Pr indicates the precision

| | | | | | WL = 1 | | | | | | | | |
|---|---|---|---|---|---|---|---|---|---|---|---|---|---|
| | TP | FN | FP | TN | AUC Score | Micro-f1 | Macro-f1 | I Error | II Error | Rec Bankruptcy | Pr Bankruptcy | Rec Healthy | Pr Healthy |
| **Svm** | 42 | 63 | 742 | 2498 | 0.585 | 0.759 | 0.478 | 22.901 | 60 | 0.4 | 0.054 | 0.771 | 0.975 |
| **Ann** | 80 | 25 | 609 | 2631 | 0.861 | 0.81 | 0.547 | 18.796 | 23.81 | 0.762 | 0.116 | 0.812 | 0.991 |
| **Logistic Regression** | 85 | 20 | 969 | 2271 | 0.755 | 0.704 | 0.484 | 29.907 | 19.048 | 0.81 | 0.081 | 0.701 | 0.991 |
| **AdaBoost** | 68 | 37 | 1072 | 2168 | 0.658 | 0.668 | 0.453 | 33.086 | 35.238 | 0.648 | 0.06 | 0.669 | 0.983 |
| **Random Forest** | 83 | 22 | 1158 | 2082 | 0.717 | 0.647 | 0.451 | 35.741 | 20.952 | 0.79 | 0.067 | 0.643 | 0.99 |
| **Gradient Boosting** | 85 | 20 | 945 | 2295 | 0.759 | 0.712 | 0.488 | 29.167 | 19.048 | 0.81 | 0.083 | 0.708 | 0.991 |
| **XgBoost** | 90 | 15 | 921 | 2319 | 0.786 | 0.72 | 0.497 | 28.426 | 14.286 | 0.857 | 0.089 | 0.716 | 0.994 |

| | | | | | WL = 2 | | | | | | | | |
|---|---|---|---|---|---|---|---|---|---|---|---|---|---|
| | TP | FN | FP | TN | AUC Score | Micro-f1 | Macro-f1 | I Error | II Error | Rec Bankruptcy | Pr Bankruptcy | Rec Healthy | Pr Healthy |
| **Svm** | 50 | 53 | 701 | 2442 | 0.631 | 0.768 | 0.492 | 22.304 | 51.456 | 0.485 | 0.067 | 0.777 | 0.979 |
| **Ann** | 75 | 28 | 541 | 2602 | 0.857 | 0.825 | 0.555 | 17.213 | 27.184 | 0.728 | 0.122 | 0.828 | 0.989 |
| **Logistic Regression** | 85 | 18 | 1024 | 2119 | 0.75 | 0.679 | 0.471 | 32.58 | 17.476 | 0.825 | 0.077 | 0.674 | 0.992 |
| **AdaBoost** | 77 | 26 | 1077 | 2066 | 0.702 | 0.66 | 0.456 | 34.267 | 25.243 | 0.748 | 0.067 | 0.657 | 0.988 |
| **Random Forest** | 89 | 14 | 979 | 2164 | 0.776 | 0.694 | 0.483 | 31.149 | 13.592 | 0.864 | 0.083 | 0.689 | 0.994 |
| **Gradient Boosting** | 82 | 21 | 1070 | 2073 | 0.728 | 0.664 | 0.461 | 34.044 | 20.388 | 0.796 | 0.071 | 0.66 | 0.99 |
| **XgBoost** | 80 | 23 | 796 | 2347 | 0.762 | 0.748 | 0.507 | 25.326 | 22.33 | 0.777 | 0.091 | 0.747 | 0.99 |

| | | | | | WL = 3 | | | | | | | | |
|---|---|---|---|---|---|---|---|---|---|---|---|---|---|
| | TP | FN | FP | TN | AUC Score | Micro-f1 | Macro-f1 | I Error | II Error | Rec Bankruptcy | Pr Bankruptcy | Rec Healthy | Pr Healthy |
| **Svm** | 43 | 58 | 628 | 2378 | 0.608 | 0.779 | 0.493 | 20.892 | 57.426 | 0.426 | 0.064 | 0.791 | 0.976 |
| **Ann** | 83 | 18 | 885 | 2121 | 0.85 | 0.709 | 0.49 | 29.441 | 17.822 | 0.822 | 0.086 | 0.706 | 0.992 |
| **Logistic Regression** | 81 | 20 | 1259 | 1747 | 0.692 | 0.588 | 0.422 | 41.883 | 19.802 | 0.802 | 0.06 | 0.581 | 0.989 |
| **AdaBoost** | 74 | 27 | 1168 | 1838 | 0.672 | 0.615 | 0.432 | 38.856 | 26.733 | 0.733 | 0.06 | 0.611 | 0.986 |
| **Random Forest** | 85 | 16 | 869 | 2137 | 0.776 | 0.715 | 0.495 | 28.909 | 15.842 | 0.842 | 0.089 | 0.711 | 0.993 |
| **Gradient Boosting** | 78 | 23 | 834 | 2172 | 0.747 | 0.724 | 0.495 | 27.745 | 22.772 | 0.772 | 0.086 | 0.723 | 0.99 |
| **XgBoost** | 77 | 24 | 816 | 2190 | 0.745 | 0.73 | 0.497 | 27.146 | 23.762 | 0.762 | 0.086 | 0.729 | 0.989 |

| | | | | | WL = 4 | | | | | | | | |
|---|---|---|---|---|---|---|---|---|---|---|---|---|---|
| | TP | FN | FP | TN | AUC Score | Micro-f1 | Macro-f1 | I Error | II Error | Rec Bankruptcy | Pr Bankruptcy | Rec Healthy | Pr Healthy |
| **Svm** | 54 | 42 | 641 | 2183 | 0.668 | 0.766 | 0.501 | 22.698 | 43.75 | 0.563 | 0.078 | 0.773 | 0.981 |
| **Ann** | 80 | 16 | 711 | 2113 | 0.87 | 0.751 | 0.517 | 25.177 | 16.667 | 0.833 | 0.101 | 0.748 | 0.992 |
| **Logistic Regression** | 85 | 11 | 1373 | 1451 | 0.7 | 0.526 | 0.393 | 48.619 | 11.458 | 0.885 | 0.058 | 0.514 | 0.992 |
| **AdaBoost** | 71 | 25 | 888 | 1936 | 0.713 | 0.687 | 0.472 | 31.445 | 26.042 | 0.74 | 0.074 | 0.686 | 0.987 |
| **Random Forest** | 80 | 16 | 807 | 2017 | 0.774 | 0.718 | 0.497 | 28.576 | 16.667 | 0.833 | 0.09 | 0.714 | 0.992 |
| **Gradient Boosting** | 72 | 24 | 925 | 1899 | 0.711 | 0.675 | 0.466 | 32.755 | 25 | 0.75 | 0.072 | 0.672 | 0.988 |
| **XgBoost** | 78 | 18 | 771 | 2053 | 0.77 | 0.73 | 0.502 | 27.302 | 18.75 | 0.813 | 0.092 | 0.727 | 0.991 |

**Table 4.** *Cont.*

| | TP | FN | FP | TN | AUC Score | Micro-f1 | Macro-f1 | I Error | II Error | Rec Bankruptcy | Pr Bankruptcy | Rec Healthy | Pr Healthy |
|---|---|---|---|---|---|---|---|---|---|---|---|---|---|
| | | | | | | **WL = 5** | | | | | | | |
| Svm | 37 | 54 | 572 | 2072 | 0.595 | 0.771 | 0.487 | 21.634 | 59.341 | 0.407 | 0.061 | 0.784 | 0.975 |
| Ann | 85 | 6 | 1107 | 1537 | 0.893 | 0.593 | 0.433 | 41.868 | 6.593 | 0.934 | 0.071 | 0.581 | 0.996 |
| Logistic Regression | 82 | 9 | 1174 | 1470 | 0.729 | 0.567 | 0.417 | 44.402 | 9.89 | 0.901 | 0.065 | 0.556 | 0.994 |
| AdaBoost | 69 | 22 | 955 | 1689 | 0.699 | 0.643 | 0.45 | 36.12 | 24.176 | 0.758 | 0.067 | 0.639 | 0.987 |
| Random Forest | 75 | 16 | 651 | 1993 | 0.789 | 0.756 | 0.52 | 24.622 | 17.582 | 0.824 | 0.103 | 0.754 | 0.992 |
| Gradient Boosting | 72 | 19 | 759 | 1885 | 0.752 | 0.716 | 0.493 | 28.707 | 20.879 | 0.791 | 0.087 | 0.713 | 0.99 |
| XgBoost | 72 | 19 | 669 | 1975 | 0.769 | 0.748 | 0.512 | 25.303 | 20.879 | 0.791 | 0.097 | 0.747 | 0.99 |

**Table 5.** Overall results on the test set (2015–2018) with look-ahead models. Rec means recall and Pr means precision.

| | TP | FN | FP | TN | AUC Score | Micro-f1 | Macro-f1 | I Error | II Error | Rec Bankruptcy | Pr Bankruptcy | Rec Healthy | Pr Healthy |
|---|---|---|---|---|---|---|---|---|---|---|---|---|---|
| | | | | | | **WL = 2** | | | | | | | |
| Svm | 51 | 52 | 759 | 2384 | 0.627 | 0.75 | 0.483 | 24.149 | 50.485 | 0.495 | 0.063 | 0.759 | 0.979 |
| Ann | 74 | 29 | 842 | 2301 | 0.819 | 0.732 | 0.493 | 26.79 | 28.155 | 0.718 | 0.081 | 0.732 | 0.988 |
| Logistic Regression | 85 | 18 | 1082 | 2061 | 0.74 | 0.661 | 0.462 | 34.426 | 17.476 | 0.825 | 0.073 | 0.656 | 0.991 |
| AdaBoost | 83 | 20 | 1042 | 2101 | 0.737 | 0.673 | 0.467 | 33.153 | 19.417 | 0.806 | 0.074 | 0.668 | 0.991 |
| Random Forest | 78 | 25 | 853 | 2290 | 0.743 | 0.73 | 0.495 | 27.14 | 24.272 | 0.757 | 0.084 | 0.729 | 0.989 |
| Gradient Boosting | 81 | 22 | 1002 | 2141 | 0.734 | 0.685 | 0.472 | 31.88 | 21.359 | 0.786 | 0.075 | 0.681 | 0.99 |
| XgBoost | 79 | 24 | 946 | 2197 | 0.733 | 0.701 | 0.48 | 30.099 | 23.301 | 0.767 | 0.077 | 0.699 | 0.989 |
| | | | | | | **WL = 3** | | | | | | | |
| Svm | 48 | 53 | 554 | 2452 | 0.645 | 0.805 | 0.513 | 18.43 | 52.475 | 0.475 | 0.08 | 0.816 | 0.979 |
| Ann | 83 | 18 | 779 | 2227 | 0.872 | 0.743 | 0.51 | 25.915 | 17.822 | 0.822 | 0.096 | 0.741 | 0.992 |
| Logistic Regression | 90 | 11 | 1136 | 1870 | 0.757 | 0.631 | 0.45 | 37.791 | 10.891 | 0.891 | 0.073 | 0.622 | 0.994 |
| AdaBoost | 87 | 14 | 872 | 2134 | 0.786 | 0.715 | 0.496 | 29.009 | 13.861 | 0.861 | 0.091 | 0.71 | 0.993 |
| Random Forest | 85 | 16 | 781 | 2225 | 0.791 | 0.743 | 0.512 | 25.981 | 15.842 | 0.842 | 0.098 | 0.74 | 0.993 |
| Gradient Boosting | 76 | 25 | 962 | 2044 | 0.716 | 0.682 | 0.469 | 32.003 | 24.752 | 0.752 | 0.073 | 0.68 | 0.988 |
| XgBoost | 83 | 18 | 841 | 2165 | 0.771 | 0.724 | 0.498 | 27.977 | 17.822 | 0.822 | 0.09 | 0.72 | 0.992 |

**Table 5.** *Cont.*

| | | | | | | | | | | WL = 4 | | | | |
|---|---|---|---|---|---|---|---|---|---|---|---|---|---|---|
| | TP | FN | FP | TN | AUC Score | Micro-f1 | Macro-f1 | I Error | II Error | Rec Bankruptcy | Pr Bankruptcy | Rec Healthy | Pr Healthy |
| **Svm** | 46 | 50 | 750 | 2074 | 0.607 | 0.726 | 0.471 | 26.558 | 52.083 | 0.479 | 0.058 | 0.734 | 0.976 |
| **Ann** | 69 | 27 | 705 | 2119 | 0.811 | 0.749 | 0.506 | 24.965 | 28.125 | 0.719 | 0.089 | 0.75 | 0.987 |
| **Logistic Regression** | 77 | 19 | 742 | 2082 | 0.77 | 0.739 | 0.507 | 26.275 | 19.792 | 0.802 | 0.094 | 0.737 | 0.991 |
| **AdaBoost** | 71 | 25 | 844 | 1980 | 0.72 | 0.702 | 0.48 | 29.887 | 26.042 | 0.74 | 0.078 | 0.701 | 0.988 |
| **Random Forest** | 81 | 15 | 770 | 2054 | 0.786 | 0.731 | 0.505 | 27.266 | 15.625 | 0.844 | 0.095 | 0.727 | 0.993 |
| **Gradient Boosting** | 79 | 17 | 880 | 1944 | 0.756 | 0.693 | 0.481 | 31.161 | 17.708 | 0.823 | 0.082 | 0.688 | 0.991 |
| **XgBoost** | 77 | 19 | 824 | 2000 | 0.755 | 0.711 | 0.49 | 29.178 | 19.792 | 0.802 | 0.085 | 0.708 | 0.991 |
| | | | | | | | | | | WL = 5 | | | | |
| | TP | FN | FP | TN | AUC Score | Micro-f1 | Macro-f1 | I Error | II Error | Rec Bankruptcy | Pr Bankruptcy | Rec Healthy | Pr Healthy |
| **Svm** | 62 | 29 | 539 | 2105 | 0.739 | 0.792 | 0.53 | 20.386 | 31.868 | 0.681 | 0.103 | 0.796 | 0.986 |
| **Ann** | 85 | 6 | 1270 | 1374 | 0.862 | 0.533 | 0.4 | 48.033 | 6.593 | 0.934 | 0.063 | 0.52 | 0.996 |
| **Logistic Regression** | 78 | 13 | 849 | 1795 | 0.768 | 0.685 | 0.48 | 32.11 | 14.286 | 0.857 | 0.084 | 0.679 | 0.993 |
| **AdaBoost** | 74 | 17 | 953 | 1691 | 0.726 | 0.645 | 0.455 | 36.044 | 18.681 | 0.813 | 0.072 | 0.64 | 0.99 |
| **Random Forest** | 79 | 12 | 715 | 1929 | 0.799 | 0.734 | 0.51 | 27.042 | 13.187 | 0.868 | 0.099 | 0.73 | 0.994 |
| **Gradient Boosting** | 74 | 17 | 866 | 1778 | 0.743 | 0.677 | 0.472 | 32.753 | 18.681 | 0.813 | 0.079 | 0.672 | 0.991 |
| **XgBoost** | 73 | 18 | 610 | 2034 | 0.786 | 0.77 | 0.527 | 23.071 | 19.78 | 0.802 | 0.107 | 0.769 | 0.991 |

### 8.1. The Most Suitable Metric

The first question we aim to answer with the experiments is the following: Does a high value of AUC mean that the model is better at predicting bankruptcy or survival probability over time? There is not a unique and simple answer. We compared the overall performance of the models using the AUC and the micro-$F_1$ score, and the two metrics present similar results both with high values: models that exhibit a high AUC also have a high micro-$F_1$ score. This direct proportionality is also evident with the macro-$F_1$ score, which we remind readers is the arithmetic average of the simple $F_1$ scores on the two classes, but the value is usually definitely lower.

The reason why the macro-$F_1$ score is much lower is related to the low precision that the models achieve in the bankruptcy class. All the best models selected with the highest value of AUC exhibit a high recall and a low precision for the bankruptcy class and this is not evident looking only at the AUC value that often is reported as something similar to the accuracy. The precision–recall trade-off is well known in the literature: there is an inverse proportionality between the two metrics since precision considers false positives and recall considers false negatives. If the model is good at predicting true positives with a low value of false negatives it will often predict negative samples as positive (High False Positive and so low precision). Moreover, in our dataset and in most of the bankruptcy datasets, the class imbalance is usually very pronounced, with an average of 97% of healthy (negative) samples in the test set versus a small 3% of bankruptcy (positive) samples. For this reason, we can assert that the precision over the bankruptcy class strongly depends on the absolute value of the negative examples in the test set. The reader may observe this dependence by looking at the confusion matrices reported in the two tables.

At the same time, this condition can be considered optimal for some financial stakeholders: If the model has a high recall on the bankruptcy class it will make some wrong predictions on healthy companies but ensure that most of the risky companies are detected and avoided in their investments.

The type II error is the other metric mostly related to the recall of the bankruptcy class, but it considers the ratio of false negatives (default companies that have been wrongly classified as healthy): to minimize this value, it is important to reduce the number of false negatives while for the recall we want to maximize the number of true positives. However, these two metrics should always be highlighted along with the AUC because they provide concrete insight into the ability of the model to correctly identify companies that are going to face financial troubles. Indeed, a higher value of AUC is due to the high precision and recall that the models achieve in the healthy (negative) class since it is over-represented.

The final user, particularly if it is a regulatory body that is responsible for monitoring the status of a company, should not blindly trust the AUC value since all the models selected on this basis may produce several false alarms (low precision for the bankruptcy class, high values of false positives). In this case, they should design and compare the ML models using the macro-$F_1$ score and then the AUC.

### 8.2. Best Model and Temporal Windows

In light of our results, we can assert that Neural Networks should be preferred among all the ML models for the bankruptcy prediction tasks since they present the best ability to generalize on new and unseen cases with every metric adopted. However, the training and design of ANNs is usually harder and requests a larger computation time, higher costs, and more experiments with respect to the other models. When computation time and costs are constraints to be kept in account or when the model should be used in high dynamic contexts, the Random Forest algorithm should be preferred since it presents almost similar performance, but it requires fewer parameters, and thus it takes less time to be designed and trained.

The temporal window analysis for tasks T1 and T2 led to the following conclusions:

- For the default prediction task (T1), the general performance increases when considering more than one year of accounting variables and this is true for both ANNs and Random

Forest. Indeed, the 5-year temporal window exhibits the best results in terms of AUC. However, in light of the discussion about the metrics, the temporal window of three years achieves the best trade-off between AUC and macro-$F_1$ score using fewer variables.

- For the survival probability prediction task (T2), the best performance is achieved when trying to predict the company status three years in advance (LAW = 3) with an AUC = 0.87 with the ANN. It should be highlighted that ANN reached for LAW = 5 a considerable AUC = 0.86. However, all the models exhibit a really low precision on the bankruptcy class except for SVM and XGBoost. Indeed the best model in terms of macro-$F_1$ score is XGBoost (LAW = 5)

In general, learning from temporal variables seems to lead to better performance, especially when the model learning capacity can learn complex patterns as happens with Neural Networks.

## 9. Conclusions

In this research work, we deeply investigated the performance of several machine-learning techniques concerning predicting bankruptcy in the American stock market. We compared the models over two different tasks: (a) default prediction using time series accounting data; (b) survival probability prediction. We performed the tasks using a dataset with 8262 companies in the period between 1999 and 2018. The dataset is also one of the contributions of this paper since it has been publicly released. We used temporal criterion to divide the dataset into training, validation, and test sets. For both tasks, Neural Networks achieve the absolute best results despite exhibiting a great variance in their results, leading to the conclusion that these models can be superior only when opportunely designed and trained with higher computational costs. Finally, we critically discuss the general use of the Area Under the Curve as a common metric to evaluate bankruptcy prediction tasks since in most cases computing precision, recall for the single classes, and the macro-$F_1$ score would better define the models' performance. Moreover, we highlighted that using more fiscal years for the prediction can improve the performance for both tasks, as has been proved in the past only for small datasets. In light of this, future works are to be related to the use of Recurrent Neural Networks and attention-models to better exploit the time-series information, considering the possible trade-off of using deep learning models with such short time series and relatively small datasets. Moreover, bankruptcy prediction could also be evaluated with unsupervised models like Isolation Tree and anomaly detection models. The current dataset could also help in that case. Finally, future works are to be related to the possible limitations of this research work. The main issue to be deeply investigated is related to the temporal dimension of the study: we were able to collect reliable data until 2018 and testing on previously unseen example has been promising. However, it would be interesting to evaluate if the current models could also generalize on different economic situations like the ones that come up with the COVID-19 pandemic and the resource crisis. Another limitation of this work is related to the class imbalance. Several techniques of effective sampling methods should be considered in the study in order to evaluate a balanced scenario as long as there is synthetic data generation.

**Author Contributions:** Conceptualization, G.L.; Methodology, G.A.; Software, M.P.; Supervision, P.M.P. and A.P.; Writing—review & editing, S.C. All authors have read and agreed to the published version of the manuscript.

**Funding:** This research received no external funding.

**Institutional Review Board Statement:** Not applicable.

**Informed Consent Statement:** Not applicable.

**Data Availability Statement:** Not applicable.

**Conflicts of Interest:** The authors declare no conflict of interest.

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
