# Peer review of "Machine Learning for Bankruptcy Prediction in the American Stock Market: Dataset and Benchmarks"

_futureinternet, doi:10.3390/fi14080244_

Round 1
Reviewer 1 Report
* The aim of the paper
This paper investigates the design and the application of different machine learning models to bankruptcy prediction tasks, e.g., a) Estimating Survival probabilities over time; b) Default prediction using time-series accounting data with different lengths.
* The main contributions
1. The first contribution is a new public bankruptcy prediction dataset to the research community, which would inspire follow-up studies.
2. The second contribution is the machine learning solutions for the two different bankruptcy prediction tasks.
* The strength of this study
1. The dataset is new and very valuable for the research community since it is publicly available and it would expect that more follow-up studies can be conducted on this dataset.
2. The whole manuscript has a clear structure.
3. The experiments are solid and the conclusions are convincing.
* The weakness
1. The novelty of the machine learning models is not the focus of this study and only those mature techniques are used.
-Specific comments referring to line numbers, tables or figures.
1. The title should be "Machine Learning for Bankruptcy Prediction in the American Stock Market: Dataset and Benchmarks", in Page 1.
2. The period "." is missing in Line 60 Page 2.
3. For the third contribution in Line 61 Page 2, explain "the most interesting metrics" so that the readers can have a better understanding.
4. A paper structure paragraph should be added in the end of the Introduction section, after Line 62 in Page 2.
5. As indicated in Table 8 Page 6, only factors for the company itself are listed and no macroeconomics variables are involved, which are also important factors for bankruptcy prediction from my humble knowledge.
6. As indicated in Table 1 Page 5, this is a highly imbalanced problem and why not consider the formulation as an anomaly detection problem and use the corresponding tools, e.g., isolation forest?
Reviewer 2 Report
I would like to thank the authors for the very interesting paper. It is clear, well written and structured, so in all a pleasure to read. There are a few minor corrections that I have found needs to be done, and one slightly larger revision that I think should be considered.
In line 20 the authors should change the way the references are printed. The submitted paper has “[1],[2]”, and it should be “[1,2]”. At line 32 “are cannot” is grammatically wrong and should be corrected. In line 84 “models class” should be corrected in “models’ class” (if my interpretation is correct).
I appreciated the extensive model validation that the authors have done, averaging the results over 100 datasets created with downsampling of the healthy population.
The only issue I found in the paper is the comparison of models by using the best results in Figure 7 for example. This should be avoided and, in my opinion, eliminated. It is true that the ANNs have can reach a higher AUC value, but that is only in a specific case (for a specific dataset) and thus is not representative at all. Therefore, I think that the analysis done in Figure 7 (and in Figure 9) is not informative and do not support the conclusions of the paper. I would remove those.
Note that normally a 3rd dataset is used to check for overfitting during the hyper-parameter phase (for example with Neural Networks) and in this case is not necessary. The analysis presented in Figure 6 and 8 is more than enough to draw all the conclusions of the paper (and support them). Additionally, I would appreciate if the authors would give, maybe in a table, also the values for the variances. At the moment one can only get an intuition of the values by looking at the figures, and it would help to have some numbers from their experiments. I think the authors should add a table that contains the numbers that are reflected in Figure 6 and 8.
I would also reconsider the first 3 lines of section 8.2, since this conclusion is not supported by the results in my opinion (the ANN can get higher values of the AUC, but only in very specific cases, and in general their AUC variance is much higher, thus making this kind of models not useful due to their low power of generalization).
In theory, to compare models a more “statistical” approach should be used (I am talking about hypothesis testing), by doing, for example, a t-test between the models’ results (assuming the distributions are Gaussian, something the authors should check). But I leave to the authors if they want to do this kind of analysis (as it would make the paper much longer, although much more sound statistically). As a side note, I think that if they did this kind of analysis, they would find that when they compare models with the ANNs they would probably discover that they cannot statistically say that the ANN is better due to the ANN’s results high variance.
As a last point (more a question), I was wondering if the authors plan to use more time-series specific methods to study this problem, because that could also be an interesting research work.
If the authors consider what I proposed above I would be very happy to support its publication in a revised form.
Reviewer 3 Report
The paper is titled – “Machine Learning for Bankruptcy prediction in the American stock Market: Dataset and Benchmarks”. It aligns with the scope of the special issue to which it has been submitted. The authors investigated the design and the application of different ML models to two different tasks related to default events: a) Estimating Survival probabilities over time; b) Default prediction using time-series accounting data with different lengths. They have also made the dataset available to the scientific community for further research. Finally, they have discussed some metrics as proposed benchmarks for future studies. The work seems novel which is supported by the results and associated discussions. However, the presentation of the certain parts of the paper should be improved. It is suggested that the authors make the necessary changes/updates to their paper as per the following comments:
1. For Task 1, the authors state – “We trained all the models using data between 1999-2011, and we made the first comparison using the Validation set (2012-2014) in terms of AUC” The training set comprises data that is 11 or more years old. The test set comprises data that is 8 or more years old. Please elaborate why any other recent datasets were not used and why the findings from such old data should still be considered relevant.
2. At the beginning of the literature review section, the authors briefly outline the advances in machine learning research but several papers cited here (references [7] to [14]) are not recent ones. For instance, [9] was published 7 years ago, [11] was published 8 years ago, and so on. Update the old references by recent papers in this field such as https://doi.org/10.3390/jsan10030039 and https://doi.org/10.3390/electronics11030421
2. It is good to see that the data has been made available as open-access. In this context please add a sub-section “Data Description” or similar to describe the dataset files and their characteristics.
3. The authors should explain whether their dataset complies with the FAIR (Findability, Accessibility, Interoperability, and Reusability) principles for scientific data management. For details on each of these principles please refer to this paper - https://pubmed.ncbi.nlm.nih.gov/26978244/
4. Please include a paragraph to discuss some potential applications of this dataset
5. The limitations of the study should be clearly highlighted
Round 2
Reviewer 3 Report
The authors have revised the paper as per all my comments. I do not have any additional comments at this point. I recommend the publication of the paper in its current form.